# Exploration of the social determinants of diarrhoea, rotavirus vaccine uptake, and vaccine 'fatigue' in Ethiopia, Kenya, and Malawi

Mackwellings Maganizo Phiri[1,2]*, Rahma Osman[3], Shewit Weldegebriel[4], Steven Sabola[1], Beatrice Ongadi[3‡], Catherine Beavis[5‡], Chikondi Mwendera[5‡], Deborah Nyirenda[2,6‡], on behalf The GHRG-G.I. Consortium¶

**1** Department of Microbiology, Kamuzu University of Health Sciences, Blantyre, Malawi, **2** Department of Social Science, Malawi Liverpool Wellcome Clinical Programme, Blantyre, Malawi, **3** Department of Microbiology Research, Kenya Medical Research Institute, Nairobi, Kenya, **4** Department of Paediatrics and Child Health, Addis Ababa University, College of Health Sciences, Ethiopia, **5** Department of Veterinary and Ecological Sciences, Institute of Infection, University of Liverpool, Liverpool, United Kingdom, **6** Liverpool School of Tropical Medicine, Pembroke Place, United Kingdom

¶ Membership of The GHRG-GI Consortium is provided in the Acknowledgements
☯ These authors contributed equally to this work
‡ These authors also contributed equally to this work
* mackwellingsphiri@gmail.com, mackphiri@outlook.com

## Abstract

Diarrhoea due to rotavirus remains a significant cause of child mortality in developing regions. Caregivers' perspectives on the social determinants of gastroenteritis and childhood vaccination, including the rotavirus vaccine, were explored through focus group discussions in Ethiopia (n = 6), Kenya (n = 14), and Malawi (n = 10), using a combination of thematic and framework analysis approaches. The results show that diarrhoea was perceived to be a burden in all three countries, particularly among infants, due to challenges in WASH (water, sanitation, and hygiene) infrastructures and poverty. Prevention strategies mentioned by the caregivers focussed on enhancing WASH interventions without mention of vaccination. Participants however expressed a lack of agency to address WASH at community level in informal settlements where the studies were conducted. Antibiotics were seen as "strong medications" and often purchased without prescription for treatment of diarrhoea, raising concerns on Anti-Microbial Resistance (AMR), while home remedies such as rice porridge were used for less severe diarrhoea symptoms. Perceived or experiential benefits and safety of previous vaccines promoted vaccine uptake in all three countries. On the other hand, limited understanding of vaccines, concerns about side effects, perceived excessive vaccination, mistrust of vaccines or suspicions of existing vaccines undermined individuals' willingness to vaccinate children. Our results highlight that a lack of attention to socio-determinants of poor health in contexts where there are several vaccines and yet a high incidence of vaccine-preventable diseases may undermine vaccine confidence. Multi-sectoral interventions to tackle

**Data availability statement:** All relevant data are within the paper and its Supporting information files.

**Funding:** This research was funded by the National Institute for Health and Care Research (NIHR133066) using UK aid from the UK Government to support global health research. The views expressed in this publication are those of the author(s) and not necessarily those of the NIHR or the UK government. Nigel Cunliffe is a National Institute for Health and Care Research (NIHR) Senior Investigator (NIHR203756). Nigel Cunliffe is affiliated to the NIHR Health Protection Research Unit in Gastrointestinal Infections at the University of Liverpool, a partnership with the UK Health Security Agency in collaboration with the University of Warwick. Deborah Nyirenda is funded by the Global Health Bioethics Network (GHBN), a Wellcome Strategic Award (228141/Z/23/Z). The views expressed are those of the author(s) and not necessarily those of the NIHR, the Department of Health and Social Care or the UK Health Security Agency, nor of the GHBN. The funder was not involved in the study's design, data collection, analysis, decision to publish, or the preparation of the manuscript. The funder was not involved in the study's design, data collection, analysis, decision to publish, or the preparation of the manuscript.

**Competing interests:** The authors have declared that no competing interests exist.

social determinants of diarrhoea and improve community understanding of vaccines are required to improve overall community health outcomes.

## Background

Despite global declines in diarrhoea-related deaths, rotavirus remains a leading cause of mortality among children under five, particularly in developing regions such as Sub-Saharan Africa (SSA) [1,2]. SSA accounts forever 23% of global childhood diarrhoea cases [3], highlighting the urgent need for effective prevention measures. Following WHO's 2009 recommendation [4], many countries have incorporated the rotavirus vaccine into their National Immunisation Programmes (NIPs). The vaccine has proven effective in preventing severe diarrhoea and related complications. However, coverage and uptake remain suboptimal, especially in low- and middle-income countries (LMICs) [5].

Several systemic and behavioural barriers contribute to these low coverage rates. While vaccine hesitancy remains a significant global challenge [6], there are also critical gaps in vaccine introduction and scale-up. These include limited cold chain infrastructure, insufficient health workforce capacity, high procurement and delivery costs, regulatory delays, and weak disease surveillance systems [7]. These challenges can delay vaccine rollout, restrict equitable access, and undermine sustainability. The SAGE Working Group on Vaccine Hesitancy defines vaccine hesitancy as *"delay in acceptance or refusal of vaccines despite the availability of vaccination services,"* influenced by confidence, convenience, and complacency [8]. Specific concerns about the safety of rotavirus vaccines, particularly related to adverse events such as bowel disorders, continue to influence public perceptions [5,9].

Infrastructural challenges further hinder vaccine uptake in many LMICs. These include inconsistent vaccine supply, limited healthcare infrastructure, and increased workloads for healthcare workers tasked with implementing new vaccines [10,11]. Additionally, social determinants of health, such as low health literacy [12], limited healthcare access [13], and poverty [14,15], also shape vaccine-related attitudes and behaviours [16]. Evidence from Kenya and other LIMICs shows that maternal education enhances awareness of the importance of childhood vaccination and is associated with increased vaccine uptake [17,18]. Conversely, low awareness of rotavirus is linked to scepticism about vaccine efficacy [19].

Broader structural inequalities, including uneven distribution of wealth [16], mistrust in health systems, and lack of basic services like water and sanitation, can increase vaccine hesitancy [20,21]. In these settings, delay in acceptance or refusal of vaccination may reflect deeper social grievances and lack of confidence in institutions. Scholars have increasingly called for targeted research to better understand and address these challenges in SSA [9]. This study explored the social determinants of diarrhoea and vaccine uptake including rotavirus vaccine, in Ethiopia, Kenya and Malawi, with the goal of identifying strategies to strengthen vaccine acceptance and delivery. We define social determinants of health as non-medical factors that influence health outcomes such as conditions in which people live, work, and access care [22].

## Methodology

### Study setting

The study was conducted within the NIHR-funded Global Health Research Group on Gastrointestinal Infections (GHRG-GI), a multi-country, multi-disciplinary project aimed at improving health outcomes from vaccine-preventable gastrointestinal infections in Eastern (Kenya and Ethiopia) and Southern (Malawi) Africa. This paper focuses on findings from the Social Science work package which aimed to understand treatment seeking for diarrhoea among caregivers of under-five children and socio-determinants of vaccine uptake, with particular emphasis on rotavirus vaccine.

The study was implemented in densely populated urban areas with limited access to social amenities and a high burden of infectious diseases. In Malawi, the study was conducted in Bangwe Township, an informal settlement on the eastern periphery of Blantyre City in southern Malawi, with an increased prevalence of HIV/AIDS, malaria, tuberculosis [23,24], and diarrhoea [24]. Most households depend on casual labour and small-scale businesses for income [25]. In Kenya, the study was conducted within the Mukuru informal settlement of Nairobi, specifically Kwa Reuben and Kwa Njenga townships, where poor water and sanitation conditions make diarrhoeal diseases common [26]. In Ethiopia, the study was conducted in Addis Ababa, focussing on the sub-cities of Teklehaimanot and Kirkos, where urbanisation has led to shortages of clean water, overcrowding, and high rates of infectious diseases [27]. All three sites have experienced cholera or serious diarrhoeal illness outbreaks in recent decades, underscoring longstanding vulnerabilities in water, sanitation, and hygiene infrastructure [28–30].

### Study design

The project employed a qualitative approach to gain in-depth understanding of community perceptions of diarrhoea diseases, treatment seeking and perceptions of vaccination including rotavirus vaccine. Qualitative research enabled us to capture detailed accounts of caregivers' lived experiences, treatment-seeking experiences, and perceptions of vaccines, which will inform development of health interventions to address barriers to treatment-seeking for diarrhoea and improve rotavirus vaccine uptake.

### Data collection

Data for the study was collected over site-specific recruitment periods. In Malawi, recruitment took place from 01/11/2023 to 30/06/2024. In Kenya, participants were enrolled between 01/06/2023 and 30/11/2024. In Ethiopia, recruitment occurred from 01/08/2023 to 20/02/2024. To gain rich insights and capture a range of collective and diverse perspectives, we conducted Focus Group Discussions (FGDs) with participants from the three countries.

FGD participants were adult male and female caregivers (aged 18 and above) of under-five children, purposefully recruited from government health facilities' catchment areas. We identified and approached them in their homes through members of the Community Engagement and Involvement (CEI) teams working on the project. In each site, the CEI team was established through a community-led selection process to ensure there was community input on the design, implementation, and dissemination of the research. In total, 30 FGDs were conducted: Malawi (n = 10); Kenya (n = 14); and Ethiopia (n = 6) (see Table 1). The number of FGDs varied across the locations based on population density and diversity, with higher-density areas requiring more FGDs to reach saturation.

Authors in Malawi (MP, SS), Kenya (RO, BO), and Ethiopia (SW) facilitated the FGDs using a study-specific topic guide. The guide included a range of topics including parents' knowledge of diarrhoea, causes of diarrhoea, perceptions of severity, treatment-seeking, and attitudes towards childhood immunisation including rotavirus vaccine (see topic guides in Supporting file 1). Weekly debriefings and monthly meetings were also held across the three sites where facilitators discussed emerging themes and areas requiring further exploration. FGDs averaged 91 minutes (ranging from 74 to 109 minutes). All FGDs were conducted in local languages (Chichewa in Malawi, Swahili in Kenya, and Amharic in Ethiopia).

**Table 1. FGD sample.**

| Setting | # of FGDs | FGDs by Gender | | | # of participants per FGD |
|---|---|---|---|---|---|
| | | Male only | Female only | Mixed Gender | |
| Malawi | 10 | 3 | 3 | 4 | 7-11 |
| Kenya | 14 | 1 | 2 | 11 | 9-11 |
| Ethiopia | 6 | 1 | 1 | 4 | 6-10 |

## Data management and analysis

Audio recordings of the FGDs were transcribed verbatim, translated into English, and thematically analysed. Authors (MP, RS, SW) individually reviewed their country-specific transcripts for quality before importing them into NVIVO 14 software for inductive and deductive coding. They initially generated free codes such as common childhood illnesses, unclean water, poor hygiene, severity of diarrhoea, buying drugs, seeing a doctor, home remedies, benefits of childhood vaccination, and concerns with childhood vaccination. These were then organised into categories including perceived causes of diarrhoea, diarrhoea care-seeking practices, and attitudes towards childhood diarrhoea. DN consolidated the country-specific coding schemes into a single codebook, which country teams used to code the remaining data, adjusting it as new information emerged. Triangulation was achieved by involving multiple people in data analysis, comparing findings across the different groups and in all the three sites. We incorporated framework analysis to understand how the emerging issues compared or varied across the three countries.

## Ethics

This study received ethics approvals from the respective ethics review boards of all participating countries: Kamuzu University of Health Sciences (P.10/22/3790) in Malawi, Kenya Medical Research Institute's Scientific Ethical Review Unit (SERU NO: KEMRI/SERU/CMR/P00221-010–2022/4637) in Kenya, the Department of Pediatrics and University of Addis Ababa (030–23-ped) in Ethiopia, and the University of Liverpool, UK (Reference number 12443). Before conducting the FGDs, participants were provided with detailed information about the study, including its purpose, procedures, and their rights. They were encouraged to ask questions and were assured of their ability to withdraw at any time without consequences. Verbal or written consent was obtained from all participants prior to their involvement, and confidentiality measures were explained, including the limits of confidentiality within the group setting.

## Results

Thematic analysis demonstrated a complex interplay of individual and community-level determinants shaping diarrhoea risks, management, and prevention efforts. Below, we delve into these non-medical determinants of diarrhoea, illustrating how they shaped community vulnerabilities and responses to prevention measures such as rotavirus vaccination.

### Perceptions of diarrhoea – Burden and severity

Across Malawi, Kenya and Ethiopia, FGD participants consistently identified diarrhoea as highly prevalent in their communities, ranking it alongside other high-burden conditions like malaria, malnutrition, tonsillitis, and pneumonia. While its widespread prevalence made diarrhoea a significant community health concern, its severity was particularly emphasized due to its acute outcomes associated with delayed or inadequate treatment. However, vulnerability to diarrhoea-related complications was considered age-dependent, with under-five children seen as highly susceptible to problems like rapid dehydration, potentially impairing blood and oxygen circulation and causing death. Beyond immediate health risks, impacts on general well-being such as loss of appetite and inability to engage in physical activities were other aspects of diarrhoea severity viewed as affecting young children.

"A child may die within 15 minutes when they are infected with diarrhoea because the child loses a lot of water, lacks oxygen, and the flow of blood is also affected". **[FGD06, Malawi]**

"Cholera and diarrhoea are the diseases that are most burdensome in this community. This is because when a child has diarrhoea, they become weak and dehydrated very quickly. As a parent, if you don't have any knowledge on helping the child then there is a high chance of you losing them". **[FGD09, Kenya]**

"I know flu and Tonsillitis are the most common illnesses affecting our community and my family; it is sometimes severe in some children. Even though it is as common as tonsillitis, diarrhoea is also another common illness that affects the community". **[FGD01, Ethiopia]**

## WASH and poverty as social determinants exacerbating vulnerability to diarrhoea

Social determinants, particularly poor sanitation, hygiene, and limited access to clean water (WASH), were significant contributors to the burden of diarrhoea across the three sites. In Malawi, poor access to clean water and reliance on contaminated water sources, such as rivers, were linked to high rates of diarrhoea. In Kenya's Mukuru informal settlement, open sewers and inadequate sanitation facilities in areas where children play led to recurrent diarrhoea cases, especially among those under five. Similarly, in Ethiopia, overcrowded living conditions with shared toilets and limited drinking water access in villages made maintaining hygiene challenging, leaving children vulnerable to illness. These findings highlight the interconnected impact of WASH issues and broader contextual factors like poverty and overcrowding.

"Sources of clean water are very limited here. We have boreholes but they are very few and far from our houses […] one borehole serves an entire location, sometimes it breaks down. That's why many people use water from wells found along the rivers which people also use as dumping sites. That's why diarrhoea is a common disease here". **[FGD01, Malawi]**

"Mukuru informal settlement has poor water and sanitation, there are open sewers everywhere and that is still the play area for children. There is a recurrence of diarrhoea among children, especially under 5 years". **[FGD01, Kenya]**

"As we live in the villages with common toilet, there is a problem of hygiene. We have limited access to drinking water. We have seven or six households using a single toilet. This is why children are vulnerable to diseases. The major cause is absence of personal hygiene, standard way of living, and quality houses". **[FGD05, Ethiopia]**

However, misconceptions about diarrhoea causes, such as parental infections, social norm violations, teething, and evil spells, were also prevalent, impacting treatment-seeking behaviour. In addition, some participants from Malawi associated a recent, large nationwide outbreak of cholera to numerous vaccinations administered.

"There are traditional beliefs like if a child suffers from diseases such as fever or diarrhoea, especially if the diarrhoea stool is green in colour, it means their mother has 'mauka' (Chlamydia) […] Some believe it means that their mother walked into a graveyard and stepped on breast milk that another breastfeeding mother spilled […] Some believe it means either of the parents had sex with another person". **[FGD03, Malawi]**

"In our community we mostly believe that diarrhoea is caused by plastic teeth in children. When they get diarrhoea, we take them to a man in our community who rubs the teeth of the child, and it stops". **[FGD05, Kenya]**

## Lack of agency and misconceptions in diarrhoea prevention

Prevention strategies for diarrhoea across the three sites were shaped by perceived causes. For instance, participants who linked diarrhoea to inadequate WASH infrastructures emphasized preventive measures such as chlorination and

maintaining clean environments. Similarly, those attributing diarrhoea to social taboos focused on adhering to social norms for prevention.

> "In my village, the people have a schedule to clean their environment twice a week in Wereda. People around here also clean their surroundings on Sunday. This is all to protect the children from the disease". **[FGD03, Ethiopia]**

> "Keeping toilets clean is a big challenge because we share the toilets. There are some people who have plans in the plots and make sure that this week one person washes the toilet, the next week someone else washes it and they make sure the toilet is okay". **[FGD08, Kenya]**

> "I always make sure that I wash my hands after changing baby diapers, or after visiting the toilet, as well as before breastfeeding my baby […] Cleaning the toilet with water treated with chlorine. Having waste bins at home". **[FGD07, Malawi]**

While communities practiced these measures, they also experienced a lack of "agency "in effectively preventing diarrhoea, leading to feelings of helplessness and a perception that diarrhoea was beyond their control. In health, agency refers to the capacity of individuals or communities to make informed decisions and take actions to influence their own circumstances and health outcomes [31]. In the context of diarrhoea prevention, this includes the ability to identify effective preventive measures, access necessary resources, and implement actions to protect against the disease. Across the three countries, this capacity was constrained by structural and personal barriers such as inadequate WASH infrastructure, environmental conditions, and infancy. For instance, participants in Malawi identified water scarcity and reliance on unsafe water sources as major barriers to maintaining hygiene. Similarly, in Kenya and Ethiopia, poor sanitation in densely populated areas and infants' indiscriminate eating habits were cited, respectively, as significant challenges, with diarrhoea often perceived as an inevitable outcome under such conditions.

> "There are problems that you cannot prevent like poor hygiene; it is difficult for us because water is a problem here. The water from Waterboard [water company] is only available very early in the morning. If you delay, there is no water, you will fetch from a borehole. Because of the long distance to the borehole, you can only fill two pails, and that's for all your water needs in the house. That's why most people fetch water from rivers". **[FGD06, Malawi]**

> "We live near drainages, there are plots and open sewers everywhere, all the plots drain dirty water into open sewers". **[FGD02, Kenya]**

> "As long as they [children] eat, they will always be infected with diarrhoea". **[FGD04, Ethiopia]**

Although participants described various methods for preventing diarrhoea, they rarely discussed immunisation as a preventive strategy. Only a small number of participants specifically identified rotavirus vaccine as a measure to prevent diarrhoea. This may be attributed to limited awareness of the vaccine, a topic we return to later when discussing determinants of vaccine uptake.

### Pluralistic treatment-seeking pathways for diarrhoea

Treatment-seeking for childhood diarrhoea in the three countries was influenced by how symptoms were interpreted and their perceived severity. In Malawi, FGD participants indicated that caregivers often sought hospital care immediately when children exhibited signs of weakness, such as lethargy and pale eyes, associating these symptoms with potentially serious illnesses like malaria. In Kenya, caregivers sometimes initially sought treatment from private chemists for convenience but turned to health facilities if symptoms persisted or worsened.

"When a child has diarrhoea, we rush to the hospital because under-five children cannot tell what the problem is. The child also feels weak when they have diarrhoea, and we assume it could be malaria. We rush to the hospital so doctors can tell what's wrong and treat the child". **[FGD04, Malawi]**

"Mostly, we take the sick child to a health facility. But at times we take them to a nearby private chemist to get over-the-counter prescription […] and if the sickness persists, we take the child to a health facility". **[FGD04, Kenya]**

Use of home remedies for managing less severe episodes of diarrhoea was widespread across the study sites, largely influenced by logistical barriers such as long waiting times at health facilities and perceptions of poor-quality care. In Kenya, caregivers reported delays in health service delivery, such as doctors taking extended breaks, which discouraged timely hospital visits and led to reliance on alternative treatments. In Malawi, natural remedies like herbs, including aloe vera and mango tree bark, were common, though their effectiveness was acknowledged as limited for more severe illnesses like cholera. In Ethiopia, caregivers frequently employed home-based solutions such as salt-sugar solutions, rice water, or honey to treat diarrhoea, while cultural practices like applying butter to the head and neck were used for fever management.

"Time is a challenge because you might go to the hospital early with intentions of coming back early but the doctors might take tea breaks in between and go up to one hour or more during those tea breaks". **[FGD07, Kenya]**

"People use natural herbs like aloe vera, leaves from an avocado tree, or barks from a mango tree. They use these herbs and get better, but not with cholera. In the past, people tried the mango tree for cholera, but it did not work". **[FGD04, Malawi]**

"Mothers may try to treat them before going to hospitals. It can be, by treating with water-salt solution, water-sugar solution and like that. They may feed them a spoon with sugar solution. They also treat them by feeding them with rice water, after boiling the rice with water. We also feed them with honey and mineral water to treat diarrhoea. In addition to that, we put butter on their head and neck when they get high fever, which gives relief". **[FGD01, Ethiopia]**

Government-funded hospitals were favoured across all sites due to their free services. Patients in Kenya and Ethiopia paid for some services, although at lower costs. However, in all three countries, public health facilities often lacked essential medications, leading patients to purchase them from private vendors. Some participants preferred antibiotics due to their perceived effectiveness, often choosing to purchase them from private merchants for smaller and more affordable quantities. A few participants voiced concerns that incomplete antibiotic dosages could make the treatment ineffective.

"Drugs that are sold illegally are more powerful. Yes, you buy them according to your budget, but the problem is vendors don't know the right dose. Your prescription might be one tablet at a time, but you end up taking 3 […] For example, if you have malaria and want to buy LA or Bactrim, they don't know how many tablets you need. You buy 2 tablets, and you think you will be okay. You may relieve pain, but, after a short while, the illness resurfaces". **[FGD04, Malawi]**

"I go to xxx hospital because it is closer to where I live, and the services are free. [But] they do not have medication. After consultation, you are sent to the nearby chemist to buy the medication, but I prefer to go to town because it is cheaper there inclusive of transport". **[FGD01, Kenya]**

Although the participants in Kenya, Ethiopia, and Malawi discussed their experiences of using antibiotics such as Doxycycline, Amoxicillin, and Metronidazole for treating diarrhoea, they were not aware that these were "antibiotics." This lack of awareness could be due to a linguistic barrier, as the term does not have a direct translation in the local languages.

**Socio-determinants of vaccine uptake and vaccine 'fatigue'**

Our study revealed that community attitudes toward childhood vaccination are strongly influenced by past experiences with infectious diseases and the perceived risks of non-vaccination. In Ethiopia, participants highlighted that historical knowledge of severe diseases like smallpox and pertussis, coupled with education from health centres and extension workers, has reinforced the importance of vaccines. Similarly, in Kenya, concerns about the debilitating effects of diseases like polio, such as causing disability, drive parents to prioritize vaccinating their children.

> "People know from the past how harmful diseases like smallpox and 'Pertussis' were. They have learned from the harms these diseases caused. [That's why] they have been taking their children for vaccine services […] I think the community knows well about the benefit of vaccine from experience, and through health education from the health centre and the extension workers". **[FGD01, Ethiopia]**

> "People are serious about vaccines especially the polio vaccine. They are mostly afraid of their children becoming handicapped, so they ensure that they are vaccinated". **[FGD07, Kenya]**

> "Vaccines are helpful. My children have been receiving vaccines since they were born. Like the vaccines for polio and measles they are helpful because my children have never had these conditions. Those are the advantages I see when my children are vaccinated". **[FGD04, Malawi]**

Engaging trusted local stakeholders like community leaders, schoolteachers, and religious figures in educating communities also enhanced vaccine uptake. In Malawi, trust in village heads was a key motivator, for they are seen as protectors of their people – "because we know that the village head will not betray their people if there's a plot to kill people," **[FGD10, Malawi]**. Similarly, in Ethiopia, the initial reluctance of some religious leaders and their gradual endorsement of vaccination over time, improved community attitudes towards vaccines.

> "Initially, religious fathers were not encouraging people to go for vaccination. But later they started doing that […] and, in general, the attitude people have towards vaccines is better now". **[FGD04, Ethiopia]**

While communities generally supported childhood vaccination, they also expressed concerns about vaccination fatigue due to the perceived high frequency of vaccine campaigns. In Malawi, some caregivers felt overwhelmed by the number of vaccines administered in a short period, comparing it to past practices with fewer vaccines. Similarly, in Kenya, frequent campaigns, such as multiple polio vaccinations annually, led to fears about potential long-term health effects and low vaccine uptake.

> "Maybe if the government reduced the vaccines that are given to children, like how it used to be in the past where children were getting 2 vaccines. Today, it is difficult for us to let our children receive these vaccines. For instance, this September alone, children have already been vaccinated and are expected to receive another vaccine before the end of the month. That's what discourages us from vaccinating our children". **[FGD02, Malawi]**

> "Most of the community members are not willing to get the vaccine since it's done each and every time, i.e., Polio is done more than thrice a year. Hence, the community believes that it might cause other diseases in the future". **[FGD03, Kenya]**

Unwillingness to vaccinate was further fuelled by suspicions surrounding COVID-19 vaccines, with fears of a sinister agenda targeting Africans. In Malawi, participants believed COVID-19 vaccines were "satanic," aimed at reducing the population, leading to broader mistrust of all vaccines, even those previously accepted. Similarly, in Kenya, some community members associated vaccines with covert family planning schemes.

"When COVID-19 vaccines arrived, people said they were satanic (evil)to reduce the population. Now we refuse even the vaccines that we have been getting before because we think they are COVID-19 vaccines in disguise". **[FGD01, Malawi]**

"For the vaccines that are taken around in our community, there are many thoughts on that. There are some who think that they are family planning schemes and many funny theories". **[FGD07, Kenya]**

Mistrust towards vaccination was also exacerbated by perceptions of coercion in both Malawi and Kenya. In Malawi, participants described the use of childhood vaccination as a mandatory condition for accessing healthcare services, which they felt undermined their autonomy and fostered resistance. Similarly, in Kenya, caregivers reported feeling obligated to follow government and healthcare provider instructions without the opportunity for informed decision-making or refusal. These practices were seen as coercive and contributed to scepticism, with some caregivers expressing a preference for voluntary vaccination based on personal choice.

"It [vaccination] should not be a must; it should be voluntary. But when our wives visit the clinic, they are asked, "Did you receive the polio vaccine?" If they say no, they say, "For your child to receive treatment, they have to receive polio vaccine first." Such things put us at risk. Why are you forcing it on us? If the child dies, it's God's will. But we should not be forced. We should decide for ourselves if we want our children to be vaccinated". **[FGD06, Malawi]**

"In our community most parents who have children usually get their children vaccinated because the Government instructs them to do so". **[FGD09, Kenya]**

Additionally, caregivers across the three sites demonstrated limited knowledge about certain childhood vaccines, such as the rotavirus vaccine, which hindered their ability to complete vaccination schedules or fully understand the benefits. In Malawi, FGD participants indicated that caregivers often vaccinated their children to comply with instructions from health care workers rather than informed understanding, reflecting a general lack of vaccine information. Similarly, in Kenya, participants noted that most mothers were unfamiliar with rotavirus, attributing this knowledge gap to inadequate education by healthcare workers.

"Another thing is the lack of information about vaccines among women with under 5 children. If women had enough information about the vaccines that are given to children, it would be easy for them to have their children vaccinated. However, most women do not have enough information about these vaccines. They go and vaccinate their children only because they have been instructed to". **[FGD03, Malawi]**

"Three quarters of the mothers with young children do not know exactly what rotavirus is […] Not many people in the community know it because the city council hospital does not educate us on it". **[FGD07, Kenya]**

Access to and perceptions of vaccine providers were other significant determinants of vaccination uptake. In Malawi, FGD participants indicated that caregivers in remote areas often missed vaccination appointments due to long distances to health facilities and the need to prioritize earning a living over making long trips for vaccines. In Kenya, some caregivers expressed mistrust of Community Health Volunteers administering vaccines during home visits, preferring trained health professionals at health facilities due to concerns about competency.

"Children miss vaccination for two reasons: the health facility is distant from residents; and parents not having time to take children for vaccination as they must make a living". **[FGD01, Malawi]**

"There are some who refused the polio vaccine because you do not take the child to hospital rather the vaccinators come to your house". [FGD06, Kenya] "Community members perceive vaccine immunizers as people who are not medically qualified to offer the service since they are community health volunteers". **[FGD03, Kenya]**

The study highlights a multifaceted interplay of individual, social, and structural factors shaping perceptions and responses to childhood diarrhoea and vaccination across three diverse settings – Malawi, Kenya, and Ethiopia. While caregivers universally recognized diarrhoea as a severe illness requiring urgent care, contextual vulnerabilities such as poor WASH conditions, poverty, and misconceptions about its causes heightened the burden. Preventative efforts varied, with participants emphasizing WASH-related measures but demonstrating limited awareness of vaccines like rotavirus as a diarrhoea prevention strategy.

Treatment-seeking pathways were pluralistic, combining home remedies for mild cases with healthcare utilization for severe symptoms. Barriers like long waiting times, perceptions of poor-quality care, and logistical challenges to accessing health services reinforced reliance on home-based solutions and private chemists. Regarding vaccination, while caregivers valued its role in preventing severe diseases, vaccine uptake was influenced by factors such as trust in providers, logistical challenges, vaccine fatigue, and sociocultural attitudes, including fears fuelled by misinformation and coercion.

## Discussion

The findings of this study highlight the critical role of social inequalities in exacerbating the burden of diarrhoea diseases and shaping community responses to preventive measures, including vaccination, in three SSA countries.

Participants from all the three sites perceived diarrhoea as a major health risk despite the existence of preventive measures such as vaccines. Environmental factors such as poor sanitation and hygiene, coupled with limited access to clean water, were seen to significantly contribute to exposure to diarrhoea and other health risks. This is consistent with previous research findings indicating that poor sanitation and contaminated water sources are primary risk factors for diarrhoea [32]. Targeted interventions to improve these socio-determinants of poor health are essential for mitigating diarrhoea [32–34].

The role of socio determinants of poor health such as poverty in exacerbating diarrhoea and poor health outcomes has been widely documented [35,36]. Our study was conducted in informal settlements in three African countries where substandard living conditions due to poverty and lack of WASH infrastructures heighten the risk of infectious disease transmission. While improving household socio-economic conditions and WASH is crucial for reducing diarrhoea and other health threats [37,38], study participants demonstrated a lack of agency to prevent or control diarrhoea at community level.

This study further highlighted that most participants were able to recognize symptoms of diarrhoea and seek treatment. However, some local beliefs on causes of diarrhoea and perceptions of health services influenced treatment seeking decisions at household level. Some caregivers in all the three sites opted to use alternative medicine or purchased incomplete doses of antibiotics from vendors, thereby delaying access to biomedical treatment. Only a few participants were aware of the consequences of self-prescription or incomplete doses of antibiotics on AMR. This lack of awareness highlights the threat of AMR in such settings where communities are exposed to incomplete doses of antibiotics without proper diagnosis or prescriptions. More generally, these findings support previous research which shows that health literacy influences treatment seeking behaviors and improves health outcomes [39].

While most participants from all the three sites cited improvements in WASH as a preventive measure for diarrhoea, only few participants cited rotavirus vaccine as one of the preventive measures. This was probably due to lack of awareness of the specific names of numerous vaccines that were being administered to children and their functions. Participants indicated that children were given too many vaccines, making it difficult for them to learn and remember them. Additionally, understanding of the rotavirus vaccine was limited, with shortage of vernacular translation for the vaccine in any of the three countries contributing to this gap in knowledge. This lack of understanding of the functions of childhood vaccines including rotavirus vaccines in settings where there are multiple vaccines could potentially have an impact on vaccine uptake [40].

Apart from lack of awareness of the functions of vaccines, misconceptions about vaccines also adversely affect vaccination rates [41]. Our findings showed that some individuals were skeptical about vaccination due to social media

messages and beliefs that vaccines were being weaponised to reduce the population in Africa. Health education programs that address these misconceptions can help to improve knowledge about disease prevention and enhance uptake of vaccines. The introduction of the rotavirus vaccine into NIPs following the WHO's recommendation in 2009 was a critical step in combating severe diarrhoea [4]. However, the focus on vaccines without addressing broad societal factors such as poverty, WASH, and lack of access to clean water has shown to perpetuate health inequalities [42]. Our findings show that neglecting the social determinants of health may unintentionally reinforce vaccine skepticism when there are multiple vaccines administered to children and yet vaccine-preventable diseases are still health risks. Consistent with the literature, the continued high prevalence of rotavirus in SSA, where over 23% of global childhood diarrhoea cases occur [3], is linked to environmental and socioeconomic conditions [43]. This highlights an urgent imperative to address underlying social determinants of health in addition to highlighting the effectiveness of vaccines to communities.

## Limitations

The study's limitations were mostly related to linguistic challenges encountered during the facilitation of FGDs in vernacular languages in all three countries. The absence of direct translations for certain medical terms, such as "rotavirus" and "antibiotics," may have limited discussions of these concepts to participants who did not know them. We discussed this challenge as a team, consulted CEI members and healthcare workers in our respective sites for advice. This lack of precise terminology led us to discuss vaccines more generally and we used the medical term for rotavirus as it was or used the phrase diarrhoea diseases. Nevertheless, other vaccines such as polio, cholera, typhoid also lack vernacular translations in Malawi and Kenya, but they are mostly discussed using medical terms and well known by communities. Mass awareness targeting the functions of various vaccines and their schedules can help to improve community understanding of these vaccines and enhance vaccine uptake. Similarly, we maintained the medical term for 'antibiotics', explained their functions and gave examples of antibiotics when facilitating FGDs. This may have affected the depth of information collected particularly among participants who were not knowledgeable of these. Future studies should also consider developing visual aids for antibiotics and vaccines to enhance the depth of discussion.

## Conclusion

In summary, the study highlighted the socio-determinants of diarrhoea, vaccine uptake or vaccine fatigue in Ethiopia, Kenya, and Malawi. Most participants in the study sites perceived diarrhoea as a major health risk due to poor WASH infrastructures in informal settings where these studies were conducted. While most of the participants cited improvements in WASH as a preventive measure, only few participants cited vaccine as preventive measure. Poor health literacy, misconceptions about vaccines and lack of vernacular translations for rotavirus and other childhood vaccines may have shaped their responses. Most participants were aware of symptoms of diarrhoea, however local beliefs about its causes contributed to the use of home remedies, thereby delaying biomedical treatment. Self-prescription of antibiotics and buying incomplete doses of antibiotics from vendors to avoid long waiting lines at the health facilities was also discussed in all the three sites. These results show that a lack of attention to socio-determinants of diarrhoea such as WASH in contexts with high incidence of vaccine-preventable diseases may continue to perpetuate health inequalities and lead to AMR. In addition, the high incidence of vaccine-preventable diseases in settings where there are multiple vaccines administered to children undermines vaccine confidence while reinforcing negative attitudes that the vaccines are not effective but harmful. Our results highlight the necessity of multi-sectoral interventions to tackle social determinants of diarrhoea, improve community understanding of vaccines and AMR to improve community health outcomes.

## Supporting information

**S1 Data. Anonymised data set supporting the study.**
(ZIP)

## Acknowledgments

We express our gratitude to participants, Community Engagement and Involvement team members. We also acknowledge the valuable contributions of the following membership of the GHRG-GI Consortium: Chisomo Msefula (Kamuzu University of Health Sciences, Blantyre, Malawi); Daniel Asrat (Addis Ababa University, Addis Ababa, Ethiopia); Prisca Benedicto (Malawi Liverpool Wellcome Programme, Blantyre, Malawi); Christina Bronowski (University of Liverpool, Liverpool, UK); Jobiba Chinkhumba (Kamuzu University of Health Sciences, Blantyre, Malawi); Helen Clough (University of Liverpool, Liverpool, UK); Jen Cornick (University of Liverpool, Liverpool, UK and Malawi Liverpool Wellcome Programme, Blantyre, Malawi); Neil French (University of Liverpool, Liverpool, UK and Malawi Liverpool Wellcome Programme, Blantyre, Malawi); Dan Hungerford (University of Liverpool, Liverpool, UK); Khuzwayo Jere (University of Liverpool, Liverpool, UK); James Ngumo Karis (Kenya Medical Research Institute, Nairobi, Kenya); Sam Kariuki (Kenya Medical Research Institute, Nairobi, Kenya); Cecilia Mbae (Kenya Medical Research Institute, Nairobi, Kenya); Amha Mekasha (Addis Ababa University, Addis Ababa, Ethiopia); Siobhan Mor (University of Liverpool, Liverpool, UK); Edson Mwinjiwa (Malawi Liverpool Wellcome Programme, Blantyre, Malawi), Latif Ndeketa (University of Liverpool, Liverpool, UK and Malawi Liverpool Wellcome Programme, Blantyre, Malawi); Phelgona Otieno (Kenya Medical Research Institute, Nairobi, Kenya); Virginia Pitzer (Yale, New Haven, USA); Yemisrach Shumeye (Addis Ababa University, Addis Ababa, Ethiopia); Abebe Habtamu Tamire (Addis Ababa University, Addis Ababa, Ethiopia); Fred Were (Kenya Medical Research Institute, Nairobi, Kenya); Mengistu Yilma (Addis Ababa University, Addis Ababa, Ethiopia; Nigel Cunliffe[*] (University of Liverpool, Liverpool, UK: nigelc@liverpool.ac.uk)

## Author contributions

**Conceptualization:** Deborah Nyirenda.

**Formal analysis:** Mackwellings Maganizo Phiri, Rahma Osman, Shewit Weldegebriel.

**Investigation:** Mackwellings Maganizo Phiri, Rahma Osman, Shewit Weldegebriel, Steven Sabola.

**Methodology:** Deborah Nyirenda.

**Supervision:** Beatrice Ongadi, Deborah Nyirenda.

**Writing – original draft:** Mackwellings Maganizo Phiri.

**Writing – review & editing:** Rahma Osman, Shewit Weldegebriel, Steven Sabola, Beatrice Ongadi, Catherine Beavis, Chikondi Mwendera, Deborah Nyirenda.

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
