## [Decision Letter · Decision Letter 0]

18 Jun 2025

PONE-D-25-01767Exploration of the Social Determinants of Diarrhoea, Rotavirus Vaccine Uptake, and Vaccine ‘Fatigue’ in Ethiopia, Kenya, and MalawiPLOS ONE

Dear Dr. Phiri,

Thank you for submitting your manuscript to PLOS ONE. After careful consideration, we feel that it has merit but does not fully meet PLOS ONE’s publication criteria as it currently stands. Therefore, we invite you to submit a revised version of the manuscript that addresses the points raised during the review process.

We look forward to receiving your revised manuscript.

Kind regards,

David J. Diemert, M.D.

Academic Editor

PLOS ONE

Journal Requirements:

3. Please note that funding information should not appear in the Acknowledgments section or other areas of your manuscript. We will only publish funding information present in the Funding Statement section of the online submission form. Please remove any funding-related text from the manuscript. 

5. We note that you have indicated that there are restrictions to data sharing for this study. For studies involving human research participant data or other sensitive data, we encourage authors to share de-identified or anonymized data. However, when data cannot be publicly shared for ethical reasons, we allow authors to make their data sets available upon request. For information on unacceptable data access restrictions, please see http://journals.plos.org/plosone/s/data-availability#loc-unacceptable-data-access-restrictions. 

6. In this instance it seems there may be acceptable restrictions in place that prevent the public sharing of your minimal data. However, in line with our goal of ensuring long-term data availability to all interested researchers, PLOS’ Data Policy states that authors cannot be the sole named individuals responsible for ensuring data access (http://journals.plos.org/plosone/s/data-availability#loc-acceptable-data-sharing-methods).

7. PLOS requires an ORCID iD for the corresponding author in Editorial Manager on papers submitted after December 6th, 2016. Please ensure that you have an ORCID iD and that it is validated in Editorial Manager. To do this, go to ‘Update my Information’ (in the upper left-hand corner of the main menu), and click on the Fetch/Validate link next to the ORCID field. This will take you to the ORCID site and allow you to create a new iD or authenticate a pre-existing iD in Editorial Manager.

8. One of the noted authors is a group or consortium: “The GHRG-GI Consortium”

In addition to naming the author group, please list the individual authors and affiliations within this group in the acknowledgments section of your manuscript. Please also indicate clearly a lead author for this group along with a contact email address.

9. Your ethics statement should only appear in the Methods section of your manuscript. If your ethics statement is written in any section besides the Methods, please move it to the Methods section and delete it from any other section. Please ensure that your ethics statement is included in your manuscript, as the ethics statement entered into the online submission form will not be published alongside your manuscript. 

Reviewers' comments:

Reviewer's Responses to Questions

**Comments to the Author**

1. Is the manuscript technically sound, and do the data support the conclusions?

Reviewer #1: Yes

2. Has the statistical analysis been performed appropriately and rigorously? 

Reviewer #1: N/A

3. Have the authors made all data underlying the findings in their manuscript fully available?

Reviewer #1: Yes

4. Is the manuscript presented in an intelligible fashion and written in standard English?

Reviewer #1: Yes

5. Review Comments to the Author

Reviewer #1: Generally, well-written manuscript that highlights the social determinants that influence vaccine uptake and diarrhoeal illness. I have suggested few comments for the consideration of the authors in the attached document.

6. PLOS authors have the option to publish the peer review history of their article (what does this mean? ). If published, this will include your full peer review and any attached files.

**Do you want your identity to be public for this peer review?** For information about this choice, including consent withdrawal, please see our Privacy Policy .

Reviewer #1: No

---

## [Author Response · Author response to Decision Letter 1]

19 Jul 2025

Ermel Johnson, MD, MPH, PhDc

Academic Editor

PLOS ONE

19 July 2025

Subject: Rebuttal Letter for Manuscript PONE-D-25-01767

Dear Dr Ermel,

We sincerely appreciate the constructive feedback provided by the reviewers and thank you for the opportunity to revise our manuscript. Below, we provide detailed responses to each comment, addressing both reviewers’ comments and journal requirements. We believe these revisions have significantly strengthened our work.

Journal Requirements:

Comment 1: Please ensure that your manuscript meets PLOS ONE's style requirements, including those for file naming.

Response: We have double checked and ensured that our manuscript meets PLOS ONE's style requirements, including those for file naming.

Comment 2: We note that the grant information you provided in the ‘Funding Information’ and ‘Financial Disclosure’ sections do not match.

Response: Thank you for raising this. As indicated in both the ‘Financial Disclosure’ and ‘Funding Information’ sections, the study was funded by the National Institute for Health and Care Research (NIHR) under grant number NIHR133066.

In addition, authors Nigel Cunliffe and Deborah Nyirenda received further support from NIHR (grant number NIHR203756) and the Wellcome Trust (grant number 228141/Z/23/Z), respectively. We have updated the ‘Funding Information’ section to ensure it is consistent with the details provided in the ‘Financial Disclosure’ section.

Comment 3. Please note that funding information should not appear in the Acknowledgments section or other areas of your manuscript. We will only publish funding information present in the Funding Statement section of the online submission form. Please remove any funding-related text from the manuscript.

Response: Thank you. We have updated the Ackowledgement section, removing information about funding (pg 26, line 575).

Comment 4. Please note that in order to use the direct billing option the corresponding author must be affiliated with the chosen institute. Please either amend your manuscript to change the affiliation or corresponding author, or email us at plosone@plos.org with a request to remove this option.

Response: Thank you. We have emailed requesting to remove this option.

5. We note that you have indicated that there are restrictions to data sharing for this study. For studies involving human research participant data or other sensitive data, we encourage authors to share de-identified or anonymized data. However, when data cannot be publicly shared for ethical reasons, we allow authors to make their data sets available upon request. For information on unacceptable data access restrictions, please see http://journals.plos.org/plosone/s/data-availability#loc-unacceptable-data-access-restrictions.

Comment b): If there are no restrictions, please upload the minimal anonymized data set necessary to replicate your study findings to a stable, public repository and provide us with the relevant URLs, DOIs, or accession numbers. Please see http://www.bmj.com/content/340/bmj.c181.long for guidelines on how to de-identify and prepare clinical data for publication. For a list of recommended repositories, please seehttps://journals.plos.org/plosone/s/recommended-repositories. You also have the option of uploading the data as Supporting Information files, but we would recommend depositing data directly to a data repository if possible.

6. In this instance it seems there may be acceptable restrictions in place that prevent the public sharing of your minimal data. However, in line with our goal of ensuring long-term data availability to all interested researchers, PLOS’ Data Policy states that authors cannot be the sole named individuals responsible for ensuring data access (http://journals.plos.org/plosone/s/data-availability#loc-acceptable-data-sharing-methods).

Response: Thank you. All relevant data are within the manuscript. We have also uplodaed FGD transcripts as Supporting Information Files, and added a ‘Supporting Information’ section to the manuscript ( page 31, lines 723-724) as shown below. We have updated our online data availability statement accordingly.

Supporting Information

S1_Folder_Anonymised Data Set Supporting the Study

Comment 7: PLOS requires an ORCID iD for the corresponding author in Editorial Manager on papers submitted after December 6th, 2016. Please ensure that you have an ORCID iD and that it is validated in Editorial Manager. To do this, go to ‘Update my Information’ (in the upper left-hand corner of the main menu), and click on the Fetch/Validate link next to the ORCID field. This will take you to the ORCID site and allow you to create a new iD or authenticate a pre-existing iD in Editorial Manager.

Response: The corresponding author’s ORCID iD has now been validated.

Comment 8: One of the noted authors is a group or consortium: “The GHRG-GI Consortium”

In addition to naming the author group, please list the individual authors and affiliations within this group in the acknowledgments section of your manuscript. Please also indicate clearly a lead author for this group along with a contact email address.

Response: Thank you. All individual authors affiliated with the consortium have been listed, with their respective institutional affiliations indicated in brackets following their names (pages 26-27). The lead author for the consortium is Nigel Cunliffe. We have clearly marked this author with asterisks, along with affiliations and email addresses, as follows:

- Nigel Cunliffe* (University of Liverpool, Liverpool, UK; nigelc@liverpool.ac.uk)

Comment 9. Your ethics statement should only appear in the Methods section of your manuscript. If your ethics statement is written in any section besides the Methods, please move it to the Methods section and delete it from any other section. Please ensure that your ethics statement is included in your manuscript, as the ethics statement entered into the online submission form will not be published alongside your manuscript.

Response: We have double checked and confirm that our ethics statement only appears in the Methods section of the manuscript.

Comment 10. Please review your reference list to ensure that it is complete and correct. If you have cited papers that have been retracted, please include the rationale for doing so in the manuscript text, or remove these references and replace them with relevant current references. Any changes to the reference list should be mentioned in the rebuttal letter that accompanies your revised manuscript. If you need to cite a retracted article, indicate the article’s retracted status in the References list and also include a citation and full reference for the retraction notice.

Response: We updated the reference following reviewers’ recommendations to clarify which SAGE working group we were referring to and use approppriate citation. We updated the segment of the passage as “The SAGE Working Group on Vaccine Hesitancy…” and replaced the citation with one that is more relevant to the change made (page 4, line 67).

Reviewers’ Comments

Introduction

1. Line 55: I agree with the authors that vaccine hesitancy could contribute to low coverage of rotavirus vaccine, but did the authors also consider gaps in introducing or scaling up rotavirus vaccines?

Response: Thank you for raising this. We have revised this section, taking into consideration how gaps in vaccination introduction and scale-up impact coverage and uptake (page 4, lines 61-79).

2. Line 56: Please clarify the specific SAGE working group you refer to? And cite the appropriate source document.

3. Response: We updated the reference following reviewers’ recommendations to clarify which SAGE working group we were referring to and use approppriate citation. We updated the segment of the passage as “The SAGE Working Group on Vaccine Hesitancy…” and replaced the citation with one that is more relevant to the change made (page 4, line 67).

4. Lines 68-69: “For instance, a study in Kenya found that incorporating hygiene education into maternal and child health interventions increased rotavirus vaccination coverage”- What was this increase? Increased by what percentage and over what period?

Response: Thank you for this query. We meant to say that maternal education was a key determinant of childhood vaccine uptake – the study found that children whose mothers had attained at least primary education were 54% more likely to be fully immunised than those whose mothers had no formal education. We have revised this section, now reads, “Evidence from Kenya and other LIMICs shows that maternal education enhances awareness of the importance of childhood immunization and is associated with increased vaccine uptake.” (page 4, lines 85-87)

Methods

5. Line 120: On recruitment, is there a definition for hospital catchment area? What radius? Proximity or otherwise can affect care-seeking behaviour, health education, as well as influence positive health-promotive behaviours.

Response: Thank you for this important observation. While there is no universally standardised definition for a hospital catchment area, for the purposes of this study, we define it as the geographic area and population that a health facility is officially designated to serve. In our context, this typically includes populations within an approximate 15-kilometre radius and covering approximately 130,000 to 450,000 people. We acknowledge that proximity to health facilities can indeed influence care-seeking behaviour, access to health education, and the adoption of health-promoting practices, and we have considered these factors in our analysis and interpretation of the findings.

Results

6. For context, can you clarify if the communities involved in this study have had previous experience of cholera or other diarrhoeal illness outbreaks? This can influence their experience and knowledge as well knowledge of preventive vaccination.

Response: Thank you for this suggestion. We have updated the study setting section as follows: “All three sites have experienced cholera or serious diarrhoeal illness outbreaks in recent decades, underscoring longstanding vulnerabilities in water, sanitation, and hygiene infrastructure.” (page 6, lines 133-135)

7. Line 326: “…using antibiotics such as Doxycycline, Amoxicillin, and Flagyl”. Since this statement is not a direct quote, I will suggest you replace Flagyl with Metronidazole.

Response: Thank you for this suggestion. We have replaced Flagyl with Metronidazole. (page 17, line 360)

We have carefully addressed all reviewer comments to enhance the manuscript's clarity. We trust that these revisions meet the expectations of the reviewers and the journal.

Thank you once again for the opportunity to revise and resubmit our manuscript. We look forward to your feedback.

Yours sincerely,

Mackwellings Phiri

(On behalf of authors)

---

## [Editor Report · Decision Letter 1]

7 Aug 2025

Exploration of the Social Determinants of Diarrhoea, Rotavirus Vaccine Uptake, and Vaccine ‘Fatigue’ in Ethiopia, Kenya, and Malawi

PONE-D-25-01767R1

Dear Dr. Phiri,

We’re pleased to inform you that your manuscript has been judged scientifically suitable for publication and will be formally accepted for publication once it meets all outstanding technical requirements.

Kind regards,

David J. Diemert, M.D.

Academic Editor

PLOS ONE
---

## [Editor Report · Acceptance letter]

PONE-D-25-01767R1

PLOS ONE

Dear Dr. Phiri,

I'm pleased to inform you that your manuscript has been deemed suitable for publication in PLOS ONE. Congratulations! Your manuscript is now being handed over to our production team.

Kind regards,

on behalf of

Dr. David J. Diemert

Academic Editor

PLOS ONE